**Comment**

# How online studies must increase their defences against AI

Gerrit Anders, Jürgen Buder, Frank Papenmeier & Markus Huff

LLM agents can now pass as human participants, threatening the validity of online social science. We urge a shift from ad-hoc checks to multi-layered, adaptive defenses, borrowing from internet anti-bot practice, and call for cooperation across researchers, platforms, and institutions, to guard against this challenge.

Online research has transformed many areas of the social sciences. Whether participants are recruited through panel providers such as Amazon Mechanical Turk, Prolific, or Qualtrics Panels, or via social media, online forums, and university mailing lists, researchers across disciplines - including psychology, sociology, communication, and political science - now routinely reach diverse populations, collect large-scale samples rapidly, and investigate human and societal behavior outside the traditional confines of the laboratory. Yet a new and formidable class of adversaries to online studies has emerged: autonomous artificial intelligence agents. Until recently, the premise that survey respondents were genuine humans was largely unquestioned at scale.

## The new adversary

Existing safeguards in online studies were designed to handle inattentiveness or misunderstanding[1] and explored the resulting threats to data quality[2,3], and not to handle deliberately adaptive attacks. Conventional measures such as attention or instruction checks were effective when the primary concern was human error, but large language models (LLMs) can reliably parse complex instructions and may circumvent simple verification steps. Most current research platforms require minimal verification of humanness, making them comparatively vulnerable to such misuse.

State-of-the-art systems such as OpenAI's Operator and Google Gemini no longer only passively generate text on command; they can function as agentic systems, able to navigate complex websites, interpret instructions, and submit forms indistinguishable from those of genuine human respondents. Open-source projects like Auto-GPT extend these capabilities further, enabling even moderately knowledgeable users to automate virtually any web-based task – potentially including mass participation in survey studies. Whereas previous worries centered on lazy or inattentive respondents, today's LLM-based agents can produce plausible, contextually appropriate, and lengthy survey responses in seconds. Because these agents are inexpensive, untiring, and easily scaled, they introduce the possibility of adversarial survey completion, both by individual participants or even as a business model.

Critically, these new threats are not monolithic. To describe the range of behaviors that could negatively influence online data collection, we propose three escalating "levels" of adversarial attack, which serve as a classification heuristic drawn from existing usage and capabilities as well as threat modeling:

- **Level 1: Human Assistance**. Individuals use LLMs as aids to generate textual responses - affecting qualitative studies, open-ended survey items, and scenario-based tasks. While not necessarily malicious, such behavior can compromise data collection when undetected. Usage has already been reported as an issue with online studies[4].
- **Level 2: Semi-Automated Agents**. Individuals employ LLM-powered browser automation or agent tools to partially or fully complete studies, often interacting with web interfaces autonomously. This type of attack could theoretically be achieved with the out-of-the-box capabilities advertised for systems such as Operator.
- **Level 3: Scalable, Full Automation**. As a speculative scenario, sophisticated, custom agents could be explicitly developed to mass-impersonate participants, circumvent defenses, and systematically monetize large-scale survey fraud. This could be a new direction for cybercrime, and has been shown in principle[5].

Each level introduces a new scale of risk - from occasional invalid responses to the industrialized corruption of research samples.

## Lessons from the broader internet

Fortunately, the social sciences need not tackle this emerging threat alone. Almost every sector with an online presence, like social networks, e-commerce giants or news outlets, has battled, often for decades, against automated bots and adversarial actors[6,7]. In this ongoing "arms race", many strategies have emerged, from CAPTCHAs and two-factor authentication to sophisticated behavior analytics, such as monitoring cursor movement, scroll patterns, and input timing. Lessons from these fields are cautionary but hopeful: while no defense is perfect, multi-layered, adaptive approaches can make automation costly and less effective for attackers. For the social sciences, the task is not to invent from scratch, but to adapt and adopt proven approaches from related fields.

To align safeguards with the spectrum of outlined adversarial behavior, we map candidate defenses to these levels. A condensed overview of the defenses discussed is provided in Table 1.

## Against Level 1 (Human assistance)

While the risk here is individual participants using LLMs to generate free-text responses, even these seemingly simple attacks can degrade data quality, particularly for studies requiring genuine engagement or open-ended input. Mitigation includes disallowing copy-paste in response fields, monitoring for abrupt pasting behavior, and tracking window or tab switching that could indicate use of external tools. Platforms may also flag suspiciously fast or formulaic text entries. However, these controls introduce trade-offs: they may hinder accessibility – particularly for participants using assistive technologies – and can add friction to legitimate study participation.

**Table 1 | Overview of defense mechanisms for each threat level**

| Threat level | Example defenses |
| --- | --- |
| Level 1 | Copy-paste restrictions/logging, paste timing checks, tab/window focus tracking, flagging formulaic responses |
| Level 2 | HTML honeypot fields, AI-targeted visual honeypots (e.g., Pseudo-Ebbinghaus), behavioral analytics (mouse movement, scrolling, reaction times), CAPTCHAs |
| Level 3 | platform-level monitoring of submission patterns, rate limiting for participation, device fingerprints, identity verification, penetration testing/red teaming |

## Against Level 2 (Semi-automated agents)

Semi-automated attacks involve LLM-driven agents that partially or fully navigate interfaces, sometimes via HTML parsing and sometimes operating via screenshots or simulated user interactions. Consequently, defenses must be chosen with agent modality in mind. For example, "honeypot" fields - hidden from human users in the HTML but visible to bots - are effective against HTML-scraping agents, while being hidden from agents that use screenshots to guide their interaction.

A further line of defense targets systematic weaknesses of LLM-based agents. Even advanced multi-modal models struggle systematically with authentic visual perception and shape recognition[8]. This vulnerability might be exploited to develop "AI honeypots" - for example, a Pseudo-Ebbinghaus Illusion, which we have used in our lab (Fig. 1). Here, the circles are actually of different sizes, though the arrangement cues recognition of this classic illusion. Agents fall for these perceptual traps, incorrectly interpreting the visual information, while humans easily detect the trick.

Further protection mechanisms include behavioral analytics, such as monitoring for unnatural mouse movements, scrolling patterns, or implausible response time, or their combinations[9].

CAPTCHAs, while imperfect, provide an out-of-the-box box solution[10], that still substantially hinders many browser-automation tools and requires some human presence; some LLM-based agents routinely resort to requesting human help for these steps, impeding fully automated scaling.

However, behavioral analytics raise ethical and privacy concerns and should be minimized, stored separately from other user data, or ideally processed only on the fly without long-term storage.

## Against Level 3 (Adversarial automation at scale)

For scenarios involving coordinated or high-volume attacks, systemic countermeasures become relevant: platforms can use real-time bot

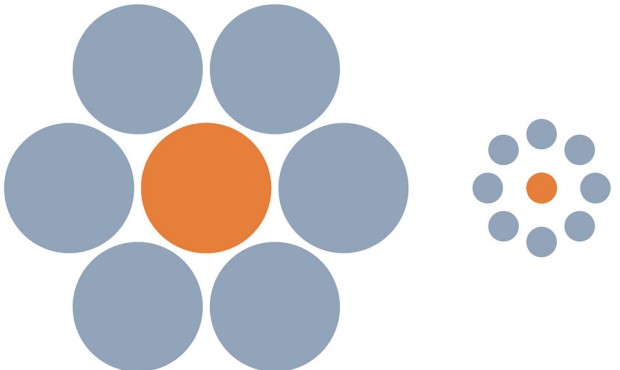

**Fig. 1 | A Pseudo-Ebbinghaus illusion as an AI-Honeypot. In contrast to the classical Ebbinghaus illusion, the inner circles have different sizes clearly visible to a human observer.** AI agents incorrectly and judge the two central circles to be of the same size, as if they detected that the display evoked an Ebbinghaus illusion.

detection, continuously updated to adapt to evolving tactics; monitor for anomalous submission patterns (e.g., unusually rapid completions, repeated IPs or device fingerprints) as well as introduce rate limiting. In addition, additional verification (e.g., via phone or even ID) could be employed and regularly tested via penetration testing. Importantly, defenders must recognize that as attackers iterate, effective protection will require defense strategies that evolve just as rapidly.

Crucially, each defense layer is imperfect on its own; together, they escalate the cost and complexity required for adversarial automation and protect research integrity.

Across all levels of attacks, individual safeguards are imperfect. Their strength lies in combination. Layered, heterogeneous defenses make automated participation more costly, less reliable, and riskier for attackers, helping to preserve research integrity. At the same time, these protections introduce trade-offs: they can add friction and reduce device compatibility making data collection more difficult and potentially introducing sampling bias. Behavioral tracking or verification requirements in addition, raise substantive privacy and fairness concerns. Researchers, therefore, need to evaluate which measures can be integrated with minimal burden (e.g., perceptual AI honey traps doubling as attention checks) and, more broadly, treat these safeguards as an ethical balancing act between robustness and privacy, rather than as a purely technical problem.

## Shared responsibility: a call for collaboration

Meeting this challenge, however, requires a fundamental change in how the field distributes responsibility. Historically, the onus for ensuring participant authenticity has fallen almost exclusively on individual researchers - who design, implement, and analyze their own studies. This decentralized model encouraged experimentation with attention checks, but it is unequal to the scale and technical complexity of today's adversarial threats. Many of the capabilities required to defend against Level 2 and 3 attacks – such as behavioral analytics or device fingerprinting are technically demanding and therefore need a platform or institute-level integration rather than being the responsibility of single researchers. This need for coordination is reinforced by the arms race character in cybersecurity in general and LLM development in particular.

Responsibility must now extend across the ecosystem: platforms like Prolific, Qualtrics, and Mechanical Turk should integrate real-time bot detection and anomaly monitoring; institutional IT and cybersecurity experts must be engaged in study design and infrastructure; panel providers and funding bodies should set standards and provide support for bot-resistant practices; and journals can raise expectations for transparent reporting of anti-bot methodologies.

In particular, journals could require that studies employ at least basic AI-directed honeypots or integrity checks and update their guidelines on reporting: authors should clearly state exclusion criteria related to suspected bot activity and disclose any additional defensive measures (e.g., rate limiting). Given how rapidly the field is evolving, such policies should emphasize minimum safeguards and researcher awareness rather than prescribing a fixed catalog of methods.

Collective, multidisciplinary action is essential for a sustainable and trustworthy research ecosystem, requiring collaboration between social scientists, computer scientists, security engineers, and platforms.

## Conclusion

Online data collection in the social sciences has entered the digital arms race. The methods that once sufficed - clever attention checks, assumed good faith could soon fall short against LLM agents. Yet, the field is not helpless. By learning from other disciplines, pooling expertise, and recognizing that shared challenges demand shared solutions, social scientists can defend against a potential new wave of adversarial attack, while at the same time reevaluating when other means of data collection, such as lab-based studies, are advantageous.

**Gerrit Anders** [1] ✉, **Jürgen Buder** [1], **Frank Papenmeier** [2] & **Markus Huff** [1,2]

[1]Leibniz-Institut für Wissensmedien, Tübingen, Germany. [2]Department of Psychology, University of Tübingen, Tübingen, Germany.
✉e-mail: g.anders@iwm-tuebingen.de

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

## Author contributions

G.A.: conceptualization, writing – original draft, writing – review & editing. J. B.: Writing – review & editing. F. P.: Writing – review & editing. M. H.: Writing – review & editing.

## Funding

## Competing interests

The authors declare no competing interests.
