## [Peer Review file · Communications Psychology]

How Online Studies Must Increase Their Defences against AI

Corresponding Author: Mr Gerrit Anders

Version 0:

Decision Letter:

**** Please ensure you delete the link to your author homepage in this e-mail if you wish to forward it to your co-authors ****

Dear Mr Anders,

Your Comment titled "Social Sciences Under Attack: Why Online Studies Must Increase Their Defences – Now" has now been seen by 2 referees, whose comments appear below. In light of their advice, I am delighted to say that we are happy, in principle, to publish a revised version in Communications Psychology.

We will not send your revised paper for further review if, in the editors' judgment, the referees' comments on the present version have been addressed. If the revised paper is in Communications Psychology format, in an accessible style, and of appropriate length, we shall accept it for publication immediately. I have attached an edited version of your manuscript, and ask you to attend to each comment in detail.

As part of the revisions I ask that you reduce the number of within-text repetitions of the threats posed by LLMs (which, as the reviewers point out, have been reviewed elsewhere) and avoid hyperbole. You will need to clearly differentiate between potential risk and evidentially widespread issues, and specify that the threats apply to internet-mediated research (online studies), not all research in the social sciences. These edits will then allow you to implement the second major requirement, which is to refocus the piece on solutions; explain these at greater level of detail; and with appropriate caveating. As for the references that the reviewers request, you will be able to accommodate these requests after adopting a referencing style more suitable to the Comment format. Especially controversial statements; facts that can be disputed; or statements that require further reading require a reference. Statements of widely appreciated points, such as the importance of online studies for social science, do not require a reference. The nature of your piece means that it does not require references that predate the advent of LLMs.

EDITORIAL REQUESTS:

* Please review the changes in the attached copy of your manuscript, which has been edited for style, and address the comments and queries I have added.

* Please check whether your manuscript contains third-party images, such as figures from the literature, stock photos, clip art or commercial satellite and map data. If any of the display items in your manuscript (figures, tables, boxes or movies) include images that are the same as, or are adaptations of, previously published images, please fill in the [Third Party Rights Table](http://www.nature.com/licenceforms/snl/thirdpartyrights-table.doc), and return to us when you submit your revised manuscript. This information will enable us to obtain the necessary rights to re-use such material. If we are unable to obtain the necessary rights to use or adapt any of the material that you wish to use, we will contact you to discuss alternative options.

* Communications Psychology uses a transparent peer review system. On author request, confidential information and data can be removed from the published reviewer reports and rebuttal letters prior to publication. If you are concerned about the release of confidential data, please let us know specifically what information you would like to have removed. Please note that we cannot incorporate redactions for any other reasons.

*If you have not done so already, please alert me to any related manuscripts from your group that are under consideration or in press at other journals, or are being written up for submission to other journals (see

www.nature.com/authors/editorial_policies/duplicate.html for details).

FORMATTING GUIDELINES:

** Length

The ideal length for Comment article in Communications Psychology is 1,500 words. We have some flexibility, however, but please ensure that your text does not exceed 1,800 words.

** Main text

Please provide three or four section headings in the main text. These should relate to the content of the article rather than being generic. Headings should be no longer than 30 characters (including spaces) and should not use punctuation.

** Figures

Please remove all figures from the main text and upload them individually, one figure per file. To ensure the swift processing of your paper please provide the highest quality, vector format, versions of your images (.ai, .eps, .psd) where available. Text and labelling should be in a separate layer to enable editing during the production process. If vector files are not available then please supply the figures in whichever format they were compiled in and not saved as flat .jpeg or .TIFF files. If your artwork contains any photographic images, please ensure these are at least 300 dpi.

* Figures should be simple and informative — multi-part figures are best avoided.

* References

References appear as superscript Arabic numerals, in order of mention. The reference list mentions references in the numerical order in which they are mentioned in the main text. If a reference is cited more than once, the same number is used throughout the text and the reference receives a single entry in the reference list.

Only papers that have been published or accepted by a named publication should be in the reference list (preprints and citations of datasets are also permitted). Unpublished/Submitted research should not be included in the reference list; it should only be mentioned briefly and parenthetically in the main text. Note that no major arguments should rely on unpublished research.

Published conference abstracts and URLs for websites should be cited parenthetically in the text, not in the reference list.

Footnotes are not used.

* Competing interests

Please include a "Competing interests" statement after the References. Note that we ask authors to declare both financial and non-financial competing interests. For more details, see <https://www.nature.com/authors/policies/competing.html>. If you have no financial or non-financial competing interests, please state so: "The authors declare no competing interests."

SUBMISSION INFORMATION:

* If you wish, you may also submit a visually arresting image, together with a concise legend, for consideration as a 'Hero Image' on our homepage. The file should be 1400x400 pixels and should be uploaded as 'Related Manuscript File'. In addition to our home page, we may also use this image (with credit) in other journal-specific promotional material.

In order to accept your paper, we require the following:

* A cover letter describing your response to our editorial requests.

* A separate document detailing your point-by-point response to any issues raised by our referees (please include the referees' comments in this document).

* The final version of your text as a Word or TeX/LaTeX file, with any tables prepared using the Table menu in Word or the table environment in TeX/LaTeX and using the 'track changes' feature in Word.

* Production-quality versions of all figures, supplied as separate files. Photographic images should be 300 dpi in RGB format (.jpg, TIFF or native Photoshop format) and any labels/scale bars included in a separate layer from the image. Line art, graphs and schemes should be vector format (.ai, .eps, .pdf); Adobe Illustrator files are preferred and will minimize production time. Any chemical structures or schemes contained within figures should additionally be supplied as separate Chemdraw (.cdx) files.

Communications Psychology is a fully open access journal. Articles are made freely accessible on publication. For further information about article processing charges, open access funding, and advice and support from Nature Research, please visit <https://www.nature.com/commpsychol/open-access>

Please note that your paper cannot be sent for typesetting to our production team until we have received this information; **therefore, please ensure that you have this ready when submitting the final version of your manuscript.**

ORCID

Communications Psychology is committed to improving transparency in authorship. As part of our efforts in this direction, we are now requesting that all authors identified as 'corresponding author' create and link their Open Researcher and Contributor Identifier (ORCID) with their account on the Manuscript Tracking System (MTS) prior to acceptance. ORCID helps the scientific community achieve unambiguous attribution of all scholarly contributions. For more information please visit <http://www.springernature.com/orcid>

For all corresponding authors listed on the manuscript, please follow the instructions in the link below to link your ORCID to your account on our MTS before submitting the final version of the manuscript. If you do not yet have an ORCID you will be able to create one in minutes.

IMPORTANT: All authors identified as 'corresponding author' on the manuscript must follow these instructions. Non-corresponding authors do not have to link their ORCIDs but are encouraged to do so. Please note that it will not be possible to add/modify ORCIDs at proof. Thus, if they wish to have their ORCID added to the paper they must also follow the above procedure prior to acceptance.

To support ORCID's aims, we only allow a single ORCID identifier to be attached to one account. If you have any issues attaching an ORCID identifier to your MTS account, please contact the [Platform Support Helpdesk](http://platformsupport.nature.com/).

Link Redacted

We hope to hear from you within two weeks; please let us know if the process may take longer.

Best regards,

Marike

Marike Schiffer, PhD
Chief Editor
Communications Psychology

REVIEWERS' COMMENTS:

Reviewer #1 (Remarks to the Author):

First of all, I would like to thank the authors for making this manuscript available to read. I think the topic is very worthwhile. As it is in a commentary/perspective-format, I will focus on comments on the literature background, the narrative introducing the problem and the framing.

1. Evidence-base:

1.1 The paper raises an important and timely concern, but the overall framing relies heavily on urgency without providing much empirical evidence. As a persuasive commentary, I think the authors can maintain a warning tone, but I would suggest anchoring key claims about the threat with referenced evidence. For example, the authors note that current LLM agents can impersonate human survey takers on a large scale. They could clarify whether this has been observed in practice, demonstrated in controlled tests or is merely a frequently discussed theoretical possibility.

1.2 Similarly, the hierarchy of the three levels is useful, but it would be helpful to clarify whether the categories are based on observed attacks, forward-looking threat modelling or a combination of both. Currently, the reader could interpret the final level as speculative without acknowledging the associated uncertainty.

1.3 I was wondering whether specific references could be provided for claims about commercial bot-detection strategies, the visual vulnerabilities of multimodal models and platform-level defences. This does not need to be exhaustive given the likely restriction on the number of references, but the authors could point to a few concrete studies and working models at least.

1.4 When discussing the Pseudo-Ebbinghaus example, the assertion that 'agents fall for these perceptual traps' (p. 4) should be supported by data or a brief description of how this has been tested; otherwise, it reads as hypothetical.

2. Further specification/explanation:

2.1 The argument sometimes jumps from possibility to inevitability. Statements such as 'capabilities accelerate [...] must transition urgently' (p. 1) are effective from a rhetorical perspective, but I would expect more nuance. I would suggest considering specifying the current limits of LLM agents, as well as their strengths.

2.2 The term 'adversary' is used rather broadly. Sometimes it refers to outright fraudsters; at other times, it refers to ordinary participants using LLMs. It would be helpful to distinguish more clearly between malicious automation and benign but complicating tool use.

2.3 The piece portrays the threat posed by bots to online studies as existential. This may be justified, but without showing where the line between a serious challenge and a paradigm shift lies, the authors risk overstating the case. It might be better to frame the threat as a spectrum of uncertainty rather than a guaranteed crisis.

2.4 The authors repeatedly draw comparisons between social science and e-commerce and other sectors. This is useful, but readers will want more detail on what makes social science distinctive. The argument is at its strongest when the authors emphasize that social science tasks are harder to automate but also harder to defend, and explain why.

2.5 Their suggestions regarding honeypots, HTML traps and behavioral analytics are sound, but they assume that most researchers understand how to implement these techniques. I would suggest adding a clarifying sentence noting that platform-level integration is more realistic than individual researchers manually adding these techniques.

3. Other perspectives:

3.1 The authors could discuss the ethical complexities of implementing stricter defences. Behavioral analytics and device fingerprints raise privacy concerns. Readers will likely expect at least a brief acknowledgement of the ethical trade-offs involved.

3.2 The proposed defences against Level 1 attacks may be too simplistic. Disallowing pasting or monitoring tab switching could interfere with accessibility requirements and generate unnecessary friction. This trade-off should be acknowledged clearly.

3.3 The authors could also discuss whether such defences risk exacerbating sampling bias by creating obstacles for lower-tech or disabled participants.

In general, I think the manuscript addresses a much needed topic and provides a relevant starting point for different actions.

Reviewer #2 (Remarks to the Author):

Although I agree with most claims made in this short paper, I doubt the significance of its contribution for two reasons:

1. The alerting tone and the form of a manifesto make the paper look like it has identified something fundamentally new, while in fact, it's just a reiteration of the trends already reported in literature, see <https://doi.org/10.1126/science.adi1778> as well as <https://doi.org/10.29115/SP-2025-0016>, <https://doi.org/10.1093/swr/svac023>, and <https://doi.org/10.1177/14707853241297009>, among others.

2. The paper tends to do overarching generalizations. This piece is relevant for the "MTurk science" only, while many social science areas stay unaffected (e.g., in-person, observational or behavioral studies) or directly benefit from the development of AI (e.g., silicon samples, agentic models, micro- and macro-simulations, lexical studies, social network analyses, media research, psychometrics). In addition, I am not sure whether the death of MTurk and similar ways of conducting social science is an unambiguously bad thing. Many psychologists, for example, noticed that MTurk modified or even hurt psychological science by increasing the amount of poorly designed self-report scales and moving away attention from the recruitment of participants (<https://doi.org/10.1177/0146167218798821>, <https://doi.org/10.1177/2515245919838781>). Moreover the alerting tone makes the paper look defensive/conservative, as if the "MTurk science" was the best thing that happened to social science (it wasn't).

I like the idea that "journals can raise expectations for transparent reporting of anti-bot methodologies" - it seems new and constructive, probably it's a promising new direction for this paper's development.

I believe that by lowering the alerting tone, acknowledging existing reports, reviews, and accounts of the same issues, and emphasizing the novel parts (such as ways to report the fraud-detecting techniques) may significantly improve the paper.

We thank the reviewers and the editor for their details and informative comments, and we appreciate their value in enhancing this comment. We addressed all comments below by dividing them into sections. Responses are indicated in blue to enhance visual clarity.

Editor

As part of the revisions I ask that you reduce the number of within-text repetitions of the threats posed by LLMs (which, as the reviewers point out, have been reviewed elsewhere) and avoid hyperbole.

Response. We removed the “Why Social Science is Uniquely Vulnerable” section as well as greatly shortened as suggested by the editor in order to remove repetition from the text. Overall, we reduced the alarmist language in the text and moved from a language of inevitability to one appropriate for a potential threat.

You will need to clearly differentiate between potential risk and evidentially widespread issues,

Response. We followed the suggestions of the reviewers (in particular the detailed suggestions by Reviewer #1) to differentiate between speculative scenarios, technological capabilities and observed issues.

and specify that the threats apply to internet-mediated research (online studies), not all research in the social sciences.

Response. We revised the comment to more precisely talk about data collection through online studies and not overgeneralize to social sciences as a whole.

These edits will then allow you to implement the second major requirement, which is to refocus the piece on solutions; explain these at greater level of detail; and with appropriate caveating.

Response. As suggested by the Editor and the reviewers we use the revision to strengthen the discussion on countermeasures. Both against particular levels as well as discussion which steps stakeholders – in particular journals – can take.

As for the references that the reviewers request, you will be able to accommodate these requests after adopting a referencing style more suitable to the Comment format. Especially controversial statements; facts that can be disputed; or statements that require further reading require a reference. Statements of widely appreciated points, such as the

importance of online studies for social science, do not require a reference. The nature of your piece means that it does not require references that predate the advent of LLMs.

Response. We removed references for common observations [1-4, 8] to accommodate literature highlighting which influence of LLMs already has been observed (Zhang, Xu & Alvero, 2025; Höhne et al., 2024) as well as an additional overview and examples for approaches to bot detection (Stein, Chen & Mangla, 2011; Xu, Liu & Li, 2020; Iliou et al., 2021).

Reviewer #1:

First of all, I would like to thank the authors for making this manuscript available to read. I think the topic is very worthwhile. As it is in a commentary/perspective-format, I will focus on comments on the literature background, the narrative introducing the problem and the framing.

1. Evidence-base:

1.1 The paper raises an important and timely concern, but the overall framing relies heavily on urgency without providing much empirical evidence. As a persuasive commentary, I think the authors can maintain a warning tone, but I would suggest anchoring key claims about the threat with referenced evidence. For example, the authors note that current LLM agents can impersonate human survey takers on a large scale. They could clarify whether this has been observed in practice, demonstrated in controlled tests or is merely a frequently discussed theoretical possibility.

Response. We thank the reviewer for pointing out that clarity and differentiation between claims and potential scenarios was missing. We generally reduced the alarming tone in the manuscript in particular in the introduction and conclusion. In addition, we addressed this by adjusting the general information in the section "The New Adversary" adding information on possibilities. We extend this for the levels by anchoring the claims with references (level 1), derivations from state-of-the-art capabilities (level 2) and highlight the speculative aspects (level 3) for each of them.

1.2 Similarly, the hierarchy of the three levels is useful, but it would be helpful to clarify whether the categories are based on observed attacks, forward-looking threat modelling or a combination of both. Currently, the reader could interpret the final level as speculative without acknowledging the associated uncertainty.

Response. We appreciate this remark and enhanced clarity regarding the levels to address this. We added a paragraph about the nature of the levels before their introduction highlighting their heuristic character. Furthermore, for each of the levels we anchor specify the (un-)certainty of each level by grounding level 1 with references,

highlighting a direct translation from capabilities for level 2 and make the speculative character of level 3 explicit to reduce ambiguity for the reader.

1.3 I was wondering whether specific references could be provided for claims about commercial bot-detection strategies, the visual vulnerabilities of multimodal models and platform-level defences. This does not need to be exhaustive given the likely restriction on the number of references, but the authors could point to a few concrete studies and working models at least.

Response. In order to strengthen the literature regarding bot detection in addition to Paul & Nikolaev (2021) we added a reference to Stein, Chen & Mangla (2011) which showcases defenses typical in large online platforms – in this example for Facebook – giving readers a concise insight into what such defenses could look like. In addition, we added a discussion on the strength and weaknesses of CAPTCHA (Xu, Liu & Li, 2020) as an important example technology, as well as a reference to Iliou et al. (2021) highlighting a working example of employing behavioral data and combining techniques. For a discussion on visual vulnerabilities, we kept Rudman et al. (2025) as a reference systematically analyzing this shape-blindness but did not extend reference in order to keep the reference list concise.

1.4 When discussing the Pseudo-Ebbinghaus example, the assertion that 'agents fall for these perceptual traps' (p. 4) should be supported by data or a brief description of how this has been tested; otherwise, it reads as hypothetical.

Response. Thank you very much for pointing out this ambiguity when presenting the Pseudo-Ebbinghaus example. We added the following paragraph in order to highlight that this is a method currently tested in our lab derived from the insight that shape recognition is systematically underdeveloped for vision-LLMs (Rudman et al., 2025):

“a Pseudo-Ebbinghaus Illusion, which we have used in our lab (Fig. 1)”

2. Further specification/explanation:

2.1 The argument sometimes jumps from possibility to inevitability. Statements such as 'capabilities accelerate [...] must transition urgently' (p. 1) are effective from a rhetorical perspective, but I would expect more nuance. I would suggest considering specifying the current limits of LLM agents, as well as their strengths.

Response. We reduced the alarming and urgent tone of the manuscript overall and added more nuance by discussing that this is an issue concerning online data collection not social science as a whole. Furthermore, we removed the specific formulation 'capabilities accelerate [...] must transition urgently'.

2.2 The term 'adversary' is used rather broadly. Sometimes it refers to outright fraudsters; at other times, it refers to ordinary participants using LLMs. It would be helpful to distinguish more clearly between malicious automation and benign but complicating tool use.

Response. We thank the reviewer for pointing out potential issues with using the term adversary this broadly. In order to guide readers in the section "The New Adversary" we added adjusted a paragraph to reflect the broad notion that adversary in this case is behavior that negatively influences data collection, but can stem from a wide range of actors:

"Critically, these new threats are not monolithic. To describe the range of behaviors that could negatively influence online data collection, we propose three escalating "levels" of adversarial attack [...]"

In addition, for each of the levels we differentiate more clearly between malicious or non-malicious issues.

2.3 The piece portrays the threat posed by bots to online studies as existential. This may be justified, but without showing where the line between a serious challenge and a paradigm shift lies, the authors risk overstating the case. It might be better to frame the threat as a spectrum of uncertainty rather than a guaranteed crisis.

Response. Thank you again for pointing out that the depiction of the threat might be overly alarmist. We adjusted the language of the manuscript and pointed out speculation more clearly when appropriate, in order to help readers differentiate between current threats and potential future issues.

2.4 The authors repeatedly draw comparisons between social science and e-commerce and other sectors. This is useful, but readers will want more detail on what makes social science distinctive. The argument is at its strongest when the authors emphasize that social science tasks are harder to automate but also harder to defend, and explain why.

Response. In order to refocus the comment as suggested by the editor and the reviewers we removed parts of the direct comparison to e-commerce and other sectors. The difficulty of needing to understand and respond to complex instructions in order to automate studies, which is the key distinction, is highlighted at the rewritten introduction to the section "the new adversary".

2.5 Their suggestions regarding honeypots, HTML traps and behavioral analytics are sound, but they assume that most researchers understand how to implement these

techniques. I would suggest adding a clarifying sentence noting that platform-level integration is more realistic than individual researchers manually adding these techniques.

Response. Thank you very much for pointing out the issue regarding technical demands of the defenses. We added a paragraph to the section “Shared Responsibility: A Call for Collaboration” addressing this issue:

“Many of the require capabilities, required to defend against Level 2 and 3 attacks – such as behavioral analytics or device fingerprinting are technically demanding and therefore need a platform or institute level integration rather than being the responsibility of single researchers. This need for coordination is reinforced by the arms race character in cybersecurity in general and LLM development in particular.”

3. Other perspectives:

3.1 The authors could discuss the ethical complexities of implementing stricter defences. Behavioral analytics and device fingerprints raise privacy concerns. Readers will likely expect at least a brief acknowledgement of the ethical trade-offs involved.

Response. Thank you for making an important point that was lacking in the manuscript. We added a paragraph regarding ethical concerns both for the behavioral analytics in particular:

“However, behavioral analytics raise ethical and privacy concerns and should be minimized, stored separately from other user data, or ideally processed only on the fly without long-term storage.”

As well as in general:

“Researchers therefore need to evaluate which measures can be integrated with minimal burden (e.g., perceptual AI honey traps doubling as attention checks) and, more broadly, treat these safeguards as an ethical balancing act between robustness and privacy, rather than a purely technical problem.”,

while also providing suggestions on how to tackle these issues.

3.2 The proposed defences against Level 1 attacks may be too simplistic. Disallowing pasting or monitoring tab switching could interfere with accessibility requirements and generate unnecessary friction. This trade-off should be acknowledged clearly.

Response. We made this tradeoff explicit in the subsection “Against Level 1 (Human Assistance):” indicating tradeoffs:

“However, these controls introduce trade-offs: they may hinder accessibility – particularly for participants using assistive technologies – and can add friction to legitimate study participation.”

In addition, we moved the previous section "Looking Forward: Adapting and Adopting Defenses" which mentioned tradeoffs closer to the discussion on mechanisms and strengthened this point.

3.3 The authors could also discuss whether such defences risk exacerbating sampling bias by creating obstacles for lower-tech or disabled participants.

Response. Thank you for pointing out friction and obstacles as a source of sampling bias. We added this point to our general discussion on defense methods:

"At the same time, these protections introduce trade-offs: they can add friction and reduce device compatibility making data collection more difficult and potentially introduce sampling bias."

In general, I think the manuscript addresses a much needed topic and provides a relevant starting point for different actions.

Reviewer #2 (Remarks to the Author):

Although I agree with most claims made in this short paper, I doubt the significance of its contribution for two reasons:

1. The alerting tone and the form of a manifesto make the paper look like it has identified something fundamentally new, while in fact, it's just a reiteration of the trends already reported in literature, see <https://doi.org/10.1126/science.adi1778> as well as <https://doi.org/10.29115/SP-2025-0016>, <https://doi.org/10.1093/swr/svac023>, and <https://doi.org/10.1177/14707853241297009>, among others.

Response. Thank you very much for the remark that the tone might be overly alarmist. We adjusted the overall tone of the manuscript in particular in introduction and discussion to guide the discussion more in the direction of a possibility and not inevitable threat.

In addition, we integrated Hühne et al. (2024) as well as Zhang, Xu and Alvero (2025) to highlight existing literature on the topic.

Furthermore, we adjusted the manuscript to extend the discussion on countermeasures to strengthen our contribution to this existing literature as suggested also by the editor.

2. The paper tends to do overarching generalizations. This piece is relevant for the "MTurk science" only, while many social science areas stay unaffected (e.g., in-person, observational or behavioral studies) or directly benefit from the development of AI (e.g.,

silicon samples, agentic models, micro- and macro-simulations, lexical studies, social network analyses, media research, psychometrics).

Response. Thank you for pointing out the unwarranted generalization in the comment. We adjusted the text to more precisely talk about online data collection as the area of tension.

In addition, I am not sure whether the death of MTurk and similar ways of conducting social science is an unambiguously bad thing. Many psychologists, for example, noticed that MTurk modified or even hurt psychological science by increasing the amount of poorly designed self-report scales and moving away attention from the recruitment of participants

(<https://doi.org/10.1177/0146167218798821>, <https://doi.org/10.1177/2515245919838781>

). Moreover the alerting tone makes the paper look defensive/conservative, as if the "MTurk science" was the best thing that happened to social science (it wasn't).

Response. Thank you very much for making the important point that this issue, even if escalated, would not be the end of social science. We adjusted the overall alerting tone enabling a more nuanced reflection of this issue by the reader. Furthermore, in the conclusion we highlighted that these issues should cause a reevaluation of researchers which issues and advantages different means of data collection have.

"[...] while at the same time reevaluating when other means of data collection, such as lab-based studies are advantageous."

I like the idea that "journals can raise expectations for transparent reporting of anti-bot methodologies" - it seems new and constructive, probably it's a promising new direction for this paper's development.

Response. We thank the reviewer for pointing out the importance of this point. In order to account for this we extended this discussion adding more explicit suggestions what role journals could play addressing this issue:

"In particular journals could require that studies employ at least basic AI-directed honeypots or integrity checks and update their guidelines on reporting: authors should clearly state exclusion criteria related to suspected bot activity and disclose any additional defensive measures (e.g., rate limiting). Given how rapidly the field is evolving, such policies should emphasize minimum safeguards and researcher awareness rather than prescribing a fixed catalog of methods."

I believe that by lowering the alerting tone, acknowledging existing reports, reviews, and accounts of the same issues, and emphasizing the novel parts (such as ways to report the fraud-detecting techniques) may significantly improve the paper.

Response. We again want to thank the reviewer for pointing out areas in which the manuscript had potential to improve. We addressed these points as discussed in the

specific points above and also increased the emphasis on the novel aspect of countermeasures.